# The association between religiosity and resilience among young trans women

**Jeremy C. Wang**[1]*, **Willi McFarland**[2,3], **Sean Arayasirikul**[4], **Erin C. Wilson**[2,3]

**1** School of Medicine, University of California, San Francisco, San Francisco, California, United States of America, **2** Department of Epidemiology & Biostatistics, University of California, San Francisco, San Francisco, California, United States of America, **3** San Francisco Department of Public Health, San Francisco, California, United States of America, **4** Department of Health, Behavior, and Society, Program in Public Health, University of California, Irvine, Irvine, California, United States of America

* Jeremy.wang@ucsf.edu

## Abstract

### Introduction

Young transgender women (trans women) experience poor health in part due to discrimination. Factors that promote resilience may help young trans women positively adapt to discrimination, resulting in attenuation of poor health outcomes. While religion is sometimes a source of stigma and transphobia, qualitative studies have identified religiosity as an important resilience resource for young trans women. The goals of this study were to quantitatively measure religiosity and resilience among young trans women and to assess whether they are associated.

### Methods

From 2012–2013, 300 young trans women between the ages of 16–24 years were enrolled in a longitudinal study; we examined the cross-sectional baseline data on demographics, religiosity, and resilience. Bivariate and multivariable logistic regression analysis examined the correlation between demographics (age, gender, race/ethnicity, education, income) and religiosity among young trans women. Additionally, bivariate and multivariable logistic regression analysis examined the association between religiosity and resilience among young trans women, controlling for age, gender, race/ethnicity, education, and income.

### Results

Participants who reported high religiosity had significantly greater odds (aOR 1.78, 95% CI 1.05–3.01, p = .03) of reporting high resilience compared to those reporting low religiosity. Black/African American participants had significantly higher odds (aOR 6.16, 95% CI 2.34–16.20, p = < .001) of reporting high religiosity compared to those who identified as White.

**Data Availability Statement:** Data cannot be shared publicly because the authors of this paper and the University of California, San Francisco (UCSF) IRB have ethical concerns related to sharing the study data publicly given the small size

of the transgender community in San Francisco. Sharing the primary data publicly may expose transgender women within the community and may place these participants at increased risk for violence and discrimination. Although de-identification of data typically renders participants unidentifiable, the small population of transgender women living in San Francisco means that even de-identified data poses a substantial risk to revealing participants' identities. Therefore, it is the view of the authors and UCSF's IRB that releasing the de-identified data publicly will place this already vulnerable population of young trans women at greater risk. Data are available from the University of California, San Francisco Office of Ethics and Compliance (contact via phone: 415-476-1825 or email: compliance@ucsf.edu) for researchers who meet the criteria for access to confidential data.

**Funding:** This study was supported by the National Institute of Mental Health (Grant # R01MH095598 https://www.nimh.nih.gov/), which was awarded to WM and EW. The funder had no role in study design, data collection and analysis, decision to publish, or preparation of the manuscript.

**Competing interests:** The authors have declared that no competing interests exist.

## Conclusion

Religiosity may be an important resilience resource for young trans women. Gender affirming religious and spiritual interventions may promote resilience among some young trans women.

## Introduction

Young transgender women (trans women) face multiple poor health outcomes due to institutional and interpersonal stigma and discrimination. Studies among young trans women have reported higher risk of experiencing physical violence, post-traumatic stress disorder, suicidal thoughts, alcohol and drug use disorders, and HIV compared to cisgender youth [1–8]. A study utilizing a nationally representative sample of transgender Americans identified that among participants, 78% experienced harassment in educational settings, 19% were refused housing, and over one-third experienced healthcare discrimination [9, 10]. Among transgender individuals, experiences with discrimination are associated with depression, asthma, and gastrointestinal diagnoses [11]. Furthermore, experiences with healthcare discrimination cause transgender individuals to avoid or delay seeking healthcare, leading to poor health outcomes [11].

Resilience has been found to be protective of poor health outcomes related to discrimination experienced by transgender people [12]. Resilience is defined as a dynamic internal and interpersonal process by which individuals positively adapt to and overcome hardship [13]. Among transgender individuals, resilience has been shown to buffer the negative effects of discrimination on mental health [14, 15].

Previously described resilience promoting factors among transgender youth include parental acceptance, school support, community connectedness, high self-esteem, and high sense of personal mastery [16–18]. These factors are theorized to boost resilience by promoting access to social support resources and individual coping mechanisms despite discriminatory treatment and adverse experiences [19].

Religiosity is an understudied potential resilience resource for transgender youth. Religiosity has been defined as the intensity, salience, importance, or centrality of religion in an individual's life [20]. Religion is a multifaceted construct that encompasses cognitive, emotional, and behavioral factors [21]. Religiosity has been theorized to include three domains–institutional (i.e., social and behavioral aspects of religion), ideological (i.e., beliefs central to a faith tradition), and personal (i.e., internalized connection to religion) [21].

In understanding the relationship between religiosity and resilience, it is also important to note that the relationship between religiosity and resilience is not necessarily unidirectional. A reciprocal relationship can exist, in which resilience enables greater civic engagement, including participation in religious practices and organizations.

Religiosity has been demonstrated to promote resilience through multiple pathways. Religiosity provides some people with coping resources such as benevolent religious reappraisal (i.e., finding positive meaning from an adverse event), coping self-efficacy (i.e., confidence in one's ability to cope with adverse events), collaborative problem solving with a higher being, and seeking spiritual support from a higher being [22, 23].

Additionally, religiosity provides some with access to religious social networks that bolster resilience. Participation in religious organizations allows individuals to develop relationships with others who share similar spiritual and religious beliefs [24]. These social connections contribute to social capital, a group-level construct that includes networks, norms, and trust,

which promotes coordinated action, and subsequently, collective access to social and material resources [25].

Social trust is a cognitive domain of social capital which is strengthened by social interactions and allows groups of people to work together for a common goal [26]. Two types of social trust are theorized to exist. Thick trust refers to trust that is built on close relationships between individuals with shared social identity (e.g., family and friends) and thin trust refers to trust that is built on looser relationships with those of different social identities [26]. Thin trust is often developed through participation in formal institutions (e.g., community organizations and work) [26].

Research with trans women of color has revealed that thick trust builds social capital within groups by allowing members to receive emotional support from each other, express themselves authentically to each other, and receive tangible support (e.g., social, physical, and financial) from one another [27]. With a strong foundation of thick trust, some trans women feel empowered to accept and embrace differences of individuals within and outside of their groups, allowing them to build greater thin trust [27]. Thin trust enables some trans women to access resources previously unfamiliar to them and to work with organizations and other groups to expand access to medical, legal, and employment resources through advocacy and political action [27].

Participation in religious communities and organizations provides individuals with access to diverse social networks that are bound together by a common spiritual identity, thus potentiating the development of thin and thick trust [28]. Ultimately, social trust fostered through participation in religious communities can facilitate development of social capital and greater resilience through increased access to social and physical resources.

Social capital has been demonstrated to bolster the resilience of trans women in a variety of settings. Trans women living with HIV have leveraged social capital within their communities to collaboratively overcome healthcare barriers, improve access to community health resources, and shift public stigma and discrimination [29]. Black/African American and Latina transgender women have reported utilizing social capital to acquire emotional support, access gender affirming medical care, and organize civic and political action [30]. These resources, made available by social capital, help trans women overcome challenges generated by societal stigma and discrimination.

Resilience and participation in religious activities can independently contribute to an individual's overall sense of wellbeing. This relationship can be understood through the "capability approach", which details that wellbeing is dependent on an individual's perceived ability to perform functions and activities that they value [31]. Inherent to the performance of most activities is some level of risk. Therefore, an individual's resilience can help them to expand their capabilities by allowing them to overcome fear and bounce back from failure in pursuit of performing desired life functions [32].

Under the framework of the "capability approach", participation in religious and spiritual activities can represent an important function among those who value religion. Therefore, freedom to participate in religious communities and organizations as an individual desires can promote wellbeing [33]. Conversely, barriers to spiritual and religious practice, which many transgender people face, may be detrimental to wellbeing if religious practice is central to their cultural identity or value system.

Among adolescents, religiosity has been found to promote resilience by providing a source of social support, facilitating adaptive coping, and providing meaning and purpose through hardship [34]. However, transgender youth's experiences with religiosity are often complicated by violence and discrimination suffered at the hands of religious institutions and individuals [35, 36]. Some religious institutions have promoted the use of "conversion therapy" for

transgender youth, resulting in massive psychological and physical damage and increasing sui-cide risk for those who are coerced into participating [37]. Furthermore, transphobic doctrine preached by some religious leaders have caused many transgender youth to internalize trans-phobia resulting in adverse mental health consequences [38]. Even in religious communities that are accepting of lesbian, gay, and queer identities, some transgender individuals report that they continue to be excluded due to their gender identity [39]. These transphobic attitudes contribute to transgender members of faith-communities experiencing profound loss of their identity and relationships, rejection, and isolation [39].

Some transgender people report that they reconcile perceived contradictions between reli-gious doctrine and their gender identity by redefining personal religious beliefs to affirm their experiences, discovering supportive faith communities, and finding peace that a higher being shaped their identity intentionally [40]. Studies show that many transgender people turn to personal religiosity and social support from affirming faith communities to cope with gender-based discrimination and stigma [39, 41, 42]. Some transgender people find their chosen fam-ily in supportive religious communities after facing rejection from their family of origin [39]. Furthermore, belief in a divine creator helps some transgender people to embrace and accept their identity [40]. In these supportive religious environments, transgender individuals report experiencing healing from past rejection and loss, inclusion, belonging, and joy [39]. Together, these studies illustrate the paradoxical impact that religion can have on the lives of transgender individuals.

While qualitative studies have extensively examined the role of religiosity on resilience of transgender people, this relationship has not been studied quantitatively among young trans women. Additionally, little is known about how religiosity varies between groups of young trans women. The primary goal of this study is to examine the association between the impor-tance of religion and resilience among young trans women to determine if greater importance of religion could be a resilience promoting factor. The secondary goal is to identify demo-graphic factors that are associated with importance of religion among young trans women. By examining these questions, the research team aims to identify a potential resilience promoting resource that can be leveraged to promote health among young trans women and to provide data regarding the subgroups for whom religiosity is most relevant.

## Methods

### Participants

This study analyses data from SHINE, a longitudinal study aimed at understanding HIV-related risk and resiliency factors among young trans women. The baseline data for SHINE was collected in the San Francisco Bay Area between August 2012 and December 2013. Partici-pants were recruited through a mixture of peer-referral, respondent driven sampling, social media outreach (e.g., Facebook, Tumblr), in-person outreach at venues frequented by young trans women (e.g., Trans March, Queer Prom), and referrals from local community organiza-tions providing social services and healthcare to trans women and youth. Additional informa-tion on recruitment can be found in a methodological publication [43]. Eligibility criteria included being assigned male sex at birth and identifying as a gender not traditionally associ-ated with male sex, being between the ages of 16 and 24 years, and living in the San Francisco Bay Area. Data for the current analysis were collected through face-to-face interviews recorded on a tablet computer. After completing the survey, participants received a $50 incentive. A total of 300 participants completed the study. All study protocols were approved by the Uni-versity of California, San Francisco Institutional Review Board. All participants provided writ-ten consent before starting the study. Participants under the age of 18 years provided written

consent in compliance with a waiver of parental consent approved by the University of California, San Francisco, Institutional Review Board.

## Measures

**Sociodemographic.** The survey collected sociodemographic information including age in years, gender [female, transgender female/trans woman, genderqueer or genderfluid, other (condensing additional sex or gender and questioning categories due to small sample size)], race/ethnicity [White, not Hispanic or Latinx; Asian, not Hispanic or Latinx; Black/African American, not Hispanic or Latinx; Hispanic or Latinx; Multiple races, not Hispanic or Latinx, other (condensing American Indian/Alaskan native, Native Hawaiian or other Pacific Islander, and other categories due to small sample size)], highest level of education received (some college or more, high school or less and currently in school, high school or less and not currently in school), and monthly income (less than or equal to $1,000, greater than $1,000).

**Independent variable.** The survey contains one item measuring religiosity, which is the independent variable for this study. Participants were asked, "How important is religion in your life?". Response options included "very important", "somewhat important", "not very important", and "not important at all". For this analysis, responses of "very important" and "somewhat important" were combined into a high religiosity category and "not very important" and "not important at all" were condensed into a low religiosity category. While various religiosity measures exist, this measure was chosen as the broad nature of the scale flexibly captures various domains of religiosity including cognitive and behavioral aspects [21].

**Outcome of interest.** The primary outcome of the study is resilience, testing the hypothesis that religiosity is associated with resilience after controlling for demographic factors. The SHINE study used the 25-item Connor Davidson resilience scale (CD-RS) to measure resilience [44]. Examples of items in this scale include, "During times of stress/crisis, I know where to turn for help" and "I give my best effort no matter what the outcome may be". Participants were asked to select how true they found each item to be for them through a four-point Likert scale ranging from 1 = rarely true to 4 = true nearly all the time (Chronbach's alpha = .88). A total resilience score with a maximum score of 100 was generated by summing each of the items. Individuals with scores greater than or equal to 75 were categorized as "high resilience" while individuals with scores less than 75 were categorized as "low resilience".

## Data analysis

Participants had the option to answer, "don't know", "refuse to answer", and "not applicable" to all questions in the survey. These responses were treated as missing values in analyses. Descriptive statistics (mean, median, SD, and 95% confidence interval for continuous variables and counts and percentages for categorical/binary variables) were calculated for the variables listed above. Bivariate and multivariable logistic regression analyses assessed associations between demographic variables and religiosity. Bivariate and multivariable logistic regression analyses controlling for demographic variables (i.e., age, gender, race/ethnicity, education, and monthly income) were further employed to analyze the association between religiosity and resilience. Odds ratios (OR), 95% confidence intervals (CI), and p-values are reported for bivariate and multivariable regression analyses. P-values below .05 were considered significant. Data analysis was conducted using STATA Statistical Software, release 16.23.

## Results

A total of 300 young trans women completed the survey. Sociodemographic characteristics of participants are reported in Table 1. The mean age of participants was 21.2 years (range: 16–

**Table 1. Sociodemographic characteristics of a sample of young trans women, San Francisco Bay Area, 2012–13 (N = 300).**

|  | Mean | SD |
|---|---|---|
| **Age**–Range: 16–24 years, Median: 22 | 21.2 | 2.17 |
|  | **N** | **%** |
| **Gender identity** |  |  |
| Female | 133 | 44.3 |
| Transgender female/trans woman | 100 | 33.3 |
| Genderqueer or genderfluid | 49 | 16.3 |
| Other | 18 | 6.0 |
| **Race/ethnicity**[a] |  |  |
| White | 107 | 35.7 |
| Asian | 18 | 6.0 |
| Black/African American | 36 | 12.0 |
| Hispanic or Latinx | 93 | 31.0 |
| Multiple races | 28 | 9.3 |
| Other | 18 | 6.0 |
| **Education** |  |  |
| High school or less and not currently in school | 96 | 32.0 |
| High school or less and currently in school | 66 | 22.0 |
| Some college or more | 138 | 46.0 |
| **Monthly income** (N = 297) |  |  |
| ≤$1,000 | 219 | 73.7 |
| >$1,000 | 78 | 26.3 |

[a]All categories other than "Hispanic or Latinx" do not include individuals who also identify as Hispanic or Latinx

24, median: 22, SD:2.17). The largest group of participants (44.3%) identified as female, 33.3% identified as transgender females/trans women, 16.3% identified as genderqueer or gender-fluid, and 6.0% identified with other genders. The largest proportion of participants (35.7%) identified their race as White, 6.0% identified as Asian, 12.0% identified as Black/African American, 31.0% identified as Hispanic or Latinx, 9.3% identified as multiple races, and 6.0% identified as other races. Nearly half (46%) completed some college or more, 22.0% completed high school or less and were currently in school, and 32.0% completed high school or less and were currently not in school. Most (73.7%) participants had a monthly income less than or equal to $1,000. Of note, White participants were more likely to have college education compared to racial/ethnic minority trans youth (61.7% vs. 37.3%, chi-square 16.47, p = 0.001) although monthly income did not differ by race/ethnicity. A slight majority (53.5%) of participants were categorized as high religiosity; 54.3% were categorized as having high resilience.

Table 2 shows results of bivariate and multivariable logistic regression analyses examining the association between demographic factors and religiosity among participants. The multivariable regression model had a sample size of 296 due to 4 (1.3%) participants missing income and/or religiosity data. In bivariate analysis, gender identity of trans women, Black/African American race/ethnicity, and having less than a high school education and not currently being in school were associated with increased religiosity. However, in the multivariable model controlling for age, gender, race/ethnicity, education, and monthly income, only individuals who identified as Black/African American had significantly higher odds of high religiosity (adjusted OR [aOR] 6.16, 95% CI 2.34–16.20, p < .001, compared to those who identified as White).

**Table 2. Bivariate and multivariable logistic regression analysis assessing associations between demographic characteristics and religiosity among young trans women, San Francisco Bay Area, 2012–13 (N = 300).**

| | Religiosity (High) | | | | | |
| | Bivariate | | | Multivariable (N = 296) | | |
| | OR | 95% CI | p[a] | aOR[b] | 95% CI | p |
|---|---|---|---|---|---|---|
| **Age (OR per year)** | .97 | .87, 1.07 | .52 | 1.02 | .89, 1.17 | .80 |
| **Gender identity** | | | | | | |
| Female | 1.00 | — | | 1.00 | — | |
| Trans woman | 1.81 | 1.07, 3.06 | **.03** | 1.41 | .78, 2.55 | .25 |
| Genderqueer or genderfluid | 1.60 | .83, 3.10 | .16 | 1.74 | .87, 3.46 | .12 |
| Other | 1.23 | .46, 3.32 | .68 | 1.39 | .50, 3.89 | .53 |
| **Race/ethnicity** | | | | | | |
| White | 1.00 | — | | 1.00 | — | |
| Asian | .66 | .22, 1.99 | .46 | 0.67 | .22, 2.04 | .48 |
| Black/African America | 5.03 | 2.85, 17.77 | **< .001** | 6.16 | 2.34, 16.20 | **< .001** |
| Hispanic, Latinx | 3.05 | .87, 2.72 | .134 | 1.26 | .68, 2.32 | .46 |
| Multiple races | 1.68 | .74, 3.98 | .21 | 1.62 | .69, 3.84 | .27 |
| Other | .64 | .50, 3.77 | .54 | 1.04 | .36, 3.03 | .94 |
| **Education** | | | | | | |
| Some college or more | 1.00 | — | | 1.00 | — | |
| High school or less, in school | 1.79 | .99, 3.24 | .06 | 1.57 | .86, 2.84 | .63 |
| High school or less, not in school | 1.80 | 1.06, 3.05 | **.03** | 1.22 | .54, 2.73 | .14 |
| **Monthly income** (N = 297) | | | | | | |
| ≤$1,000 | 1.00 | — | | 1.00 | — | |
| >$1,000 | .88 | .52, 1.48 | .63 | 0.93 | .52, 1.68 | .81 |

OR: odds ratio; aOR: adjusted odds ratio; CI: confidence interval

[a]Bolded p-values indicate that they are significant at the .05 significance level

[b]Model includes age, gender, race/ethnicity, education, and monthly income

Table 3 shows results from bivariate and multivariable logistic regression analyses assessing the association between religiosity and resilience controlling for age, gender, race/ethnicity, education, and monthly income. In the multivariable model, those who reported high religiosity had 1.78 (95% CI 1.05–3.01, p = .03) times greater odds of having high resilience compared to those who reported low religiosity after controlling for age, gender, race/ethnicity, education, and income. Black/African American (aOR 4.25, 95% CI 1.56, 11.62, p = .005) and Hispanic/Latinx race/ethnicity (aOR 2.49, 95% 1.30, 4.77, p = .006), and monthly income >$1,000 (aOR 3.87, 95% CI 2.00, 7.50, p = < .001) were also independently associated with increased resilience. Those who reported their gender identity as "other" had significantly decreased odds (aOR .29, 95% CI .09, .95, p = .04) of reporting high resilience compared to those who reported their gender identity as female.

## Discussion

We found that young trans women who reported high religiosity had a significantly greater odds of having high resilience compared to those who reported low religiosity. Our finding is consistent with qualitative research reporting that despite stigma from religious institutions, some transgender individuals identify religiosity to be a vital source of resilience [45, 46]. Specifically, transgender participants from prior research have reported that their religiosity helps

**Table 3. Bivariate and multivariable logistic regression analysis assessing the association between religiosity and resilience among young trans women controlling for demographic variables, San Francisco Bay Area, 2012–13 (N = 300).**

| | Resilience (High) | | | | | |
| | Bivariate | | | Multivariable (N = 296) | | |
| | OR | 95% CI | p[a] | aOR[b] | 95% CI | p |
|---|---|---|---|---|---|---|
| **Religiosity** | | | | | | |
| Low | 1.00 | — | | 1.00 | — | |
| High | 2.07 | 1.30, 3.29 | **.002** | 1.78 | 1.05, 3.01 | **.03** |
| **Race/ethnicity** | | | | | | |
| White | 1.00 | — | | 1.00 | — | |
| Asian | .69 | .24, 1.97 | .49 | 0.50 | .16, 1.53 | .22 |
| Black/African American | 5.71 | 2.30, 14.18 | < .001 | 4.25 | 1.56, 11.62 | **.005** |
| Hispanic, Latinx | 2.76 | 1.55, 4.91 | **.001** | 2.49 | 1.30, 4.77 | **.006** |
| Multiple races | 1.59 | .69, 3.67 | .28 | 1.21 | .49, 2.96 | .68 |
| Other | .69 | .24, 1.97 | .49 | 0.86 | .28, 2.65 | .79 |
| **Age** | .97 | .87, 1.07 | .54 | 1.03 | .89, 1.19 | .70 |
| **Gender Identity** | | | | | | |
| Female | 1.00 | — | | 1.00 | — | |
| Trans woman | 1.39 | .82, 2.35 | .22 | .75 | .40, 1.41 | .37 |
| Genderqueer or genderfluid | 1.34 | .69, 2.61 | .38 | 1.06 | .51, 2.19 | .87 |
| Other | .36 | .12, 1.06 | .06 | .29 | .09, .95 | **.04** |
| **Education** | | | | | | |
| Some college or more | 1.00 | — | | 1.00 | — | |
| High school or less and currently in school | 1.85 | 1.02, 3.39 | **.045** | 1.24 | .66, 2.35 | .49 |
| High school or less and not currently in school | 1.36 | .81, 2.30 | .25 | 2.07 | .88, 4.90 | .10 |
| **Monthly income** (N = 297) | | | | | | |
| ≤$1,000 | 1.00 | — | | 1.00 | — | |
| >$1,000 | 2.76 | 1.58, 4.84 | < .001 | 3.87 | 2.00, 7.50 | < .001 |

OR: odds ratio; aOR: adjusted odds ratio; CI: confidence interval

[a]Bolded p-values indicate that they are significant at the .05 significance level

[b]Adjusting for race/ethnicity, age, gender identity, education, and monthly income

them to cope with adversity, foster hope, accept their personal identity, find their chosen family, and heal past trauma [39–42, 46]. The present study extends the evidence base by quantitatively demonstrating the positive association between religiosity and resilience among young trans women.

We also found that Black/African American young trans women were significantly more likely than young trans women of other races to report high religiosity. A population-level study among LGBT adults in the U.S. found that Black/African American participants had the highest rates of identifying as religious compared to those of other races [47]. Religious institutions have long been central to the lives of many Black/African Americans. Black/African American churches served as safe havens for many enslaved Black/African Americans and nurtured abolitionist ideals and movements [48]. During the civil rights movement, churches and mosques became focal points of political and social action [48]. Black/African American religious institutions continue to carry out these functions today, fostering social connections and organizing political action to combat racism and discrimination [48, 49]. For many Black/African American transgender people, their relationship with religiosity may be complicated

by the cis-normative and transphobic views within some religious communities [50]. Nonetheless, religious institutions are a vital source of social support for many Black/African American LGBT individuals [51]. In past studies, LGBT-affirming Black/African American churches have been shown to serve as essential spaces of emotional healing and social connection for many Black/African American sexual minority men [52].

The San Francisco Bay Area where the study was conducted is home to several LGBT-affirming religious congregations that predominantly serve the Black/African American community and other minoritized racial and ethnic groups. It is possible that some of the young Black/African American trans women in our study accessed these religious institutions, which fostered increased religiosity. Others may have attended less-affirming congregations, where the social support, community, and spiritual support they received overshadowed some of the gender-based stigma or discrimination they may have faced in these environments. Some participants may also have maintained personal spirituality without current affiliation with formal faith-based organizations. Additional research should examine different ways in which trans youth interface with religion and how these methods may uniquely impact resilience.

Our study revealed that Black/African American and Latinx transgender youth had greater resilience than those who identified as White. This relationship is consistent with findings from past studies among cisgender populations; however, to our knowledge, no study has identified this relationship among transgender youth [53–56]. Among cisgender populations, high resilience within Black/African American and Hispanic populations are attributed to religious and spiritual practices, racial socialization and identification, and centrality of intergenerational support [53–56]. Higher resilience among Black/African American and Hispanic youth may stem from close social ties within Black/African American and Hispanic transgender communities and greater family acceptance (among Black/African American transgender individuals) than their White counterparts [57]. Furthermore, the present study demonstrates that young Black/African American trans women have higher religiosity than their counterparts, which may contribute to increased resilience. More research should be conducted to understand the reason for higher resilience among Black/African American and Hispanic compared to White young trans women.

Given the positive association between religiosity and resilience, efforts to support the spiritual needs of trans women may help to foster resilience. The importance of religion in the lives of some young trans women may stem from past or current participation in faith-based organizations. Within the United States, various LGBTQ-affirming places of worship exist. Mainstream religious denominations including the Unitarian Universalist Association, the United Church of Christ, the Episcopal church, Reform and Conservative Judaism, the Evangelical Lutheran Church in America, and the Presbyterian Church (USA) have officially adopted LGBTQ-inclusive policies [58]. Glide Memorial Church, in San Francisco, California has a long history of LGBTQ inclusion and advocacy and has hosted a Gender Expression & Identity summit, with the goal of fostering solidarity with the trans community and to hold space for trans individuals to express their spirituality [59]. Masjid al-Rabia is an LGBTQ-affirming Islamic community center in Chicago, Illinois that facilitates support groups for trans and gender nonconforming Muslims [60]. The Brooklyn Zen Center in Brooklyn, New York creates space for trans Buddhists, by hosting events that highlight the voices of trans Buddhist leaders [61]. While not comprehensive, these gender affirming faith-based organizations represent some of the religious communities that transgender people may participate in. Effort should be made to expand and support faith-based organizations like these, which serve as promoters of social capital and resilience for trans individuals. Additionally, transphobic doctrine and attitudes within all religious organizations must be reformed so that trans women can have greater access to this potential resilience promoting resource.

Our study fills an important gap in research on the connection between religiosity and resilience among young trans women. Future studies are needed to identify factors that mediate the relationship between religiosity and resilience among transgender youth (e.g., increased social capital, social support, positive outlook, well-being). Understanding these intermediate factors can help public health practitioners design interventions targeting resilience promoting mediators, regardless of religious background or interest in organized religious practices. Additionally, future studies could investigate Black/African American transgender youths' experiences within transgender-affirming religious communities and explore partnerships with these communities to promote health among young Black/African American young trans women. Public health efforts in collaboration with faith leaders may serve as a vital resilience promoting and health protecting resource for many young trans women.

## Limitations

This study contains certain limitations. First, variables were self-reported by participants, limiting their reliability. Second, this study is cross-sectional, limiting our ability to determine the temporal nature of the relationship between religiosity and resilience; namely, that being more religious causes increased resilience. It is also possible that individuals with greater resilience are more likely to participate in civil society, including in religious organizations, thus leading to increased religiosity. Third, this study was conducted in the San Francisco Bay Area, a liberal metropolitan area with a large LGBT community and many gender-affirming faith communities. Young trans women who live in communities with differing degrees of LGBT acceptance and distinct religious environments may be impacted differently by religiosity. Fourth, information on religious affiliation with respect to religion, denomination, and degree of trans-affirmation was not collected from participants. The degree of trans-affirmation within a religious organization's policies and culture may influence the effects that religiosity has on resilience of transgender members. Therefore, lack of consideration of religious affiliation of participants represents a limitation of this study. Additionally, the degree of trans-affirmation of faith communities that trans women participated in may independently influence their religiosity and resilience. Therefore, the observed association between religiosity and resilience within this study could theoretically be attributed to the confounding effects of the trans-affirmativeness of faith-based organizations that participants were a part of. Future studies may examine the relationships between religiosity and resilience while controlling for trans-affirmativeness of faith-based communities that participants are members of. Finally, our measurement of religiosity relied on participant report of the importance of religion in their life. This measurement of religion does not delineate between the institutional, ideological, and personal domains of religiosity. As a result, we are unable to attribute increased resilience to specific aspects of religiosity such as participation in religious services, faith-based belief systems, or personal spirituality. Future studies should examine the distinctive effects of different domains of religiosity on resilience among young trans women.

## Conclusions

To our knowledge, this is the first study to find that higher religiosity is significantly associated with greater resilience among young trans women. This study also identifies Black/African American racial identity as a positive correlate of religiosity among young trans women and an independent correlate of increased resilience. Findings highlight that religiosity is an important source of resilience for many young trans women. Affirming faith-based communities and organizations that support the religiosity of young trans women may help to promote resilience. Future studies are needed to investigate factors that mediate the relationship

between religiosity and resilience to identify novel strategies for promoting resilience among young trans women of varying religious experiences.

## Acknowledgments

We would like to thank all the trans women who participated in this study. Their contributions will contribute to greater understanding of the sources of immense strength and resilience withing the trans community.

## Author Contributions

**Conceptualization:** Jeremy C. Wang, Willi McFarland, Sean Arayasirikul, Erin C. Wilson.

**Data curation:** Willi McFarland, Sean Arayasirikul, Erin C. Wilson.

**Formal analysis:** Jeremy C. Wang, Erin C. Wilson.

**Funding acquisition:** Willi McFarland, Erin C. Wilson.

**Investigation:** Jeremy C. Wang, Willi McFarland, Erin C. Wilson.

**Methodology:** Jeremy C. Wang, Willi McFarland, Sean Arayasirikul, Erin C. Wilson.

**Project administration:** Willi McFarland, Sean Arayasirikul.

**Supervision:** Willi McFarland, Erin C. Wilson.

**Visualization:** Jeremy C. Wang.

**Writing – original draft:** Jeremy C. Wang.

**Writing – review & editing:** Jeremy C. Wang, Willi McFarland, Sean Arayasirikul, Erin C. Wilson.

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
