## [Decision Letter · Decision Letter 0]

19 Sep 2022

PONE-D-22-01853The association between religiosity and resilience among young trans womenPLOS ONE

Dear Dr. Wang,

Thank you for submitting your manuscript to PLOS ONE. After careful consideration, we feel that it has merit but does not fully meet PLOS ONE’s publication criteria as it currently stands. Therefore, we invite you to submit a revised version of the manuscript that addresses the points raised during the review process.

Thank you for your submission - in your revised version, please respond to each comment from each reviewer. We look forward to reading your revised submission.==============================

We look forward to receiving your revised manuscript.

Kind regards,

Amy Michelle DeBaets, PhD

Academic Editor

PLOS ONE

2.Please update your submission to use the PLOS LaTeX template. The template and more information on our requirements for LaTeX submissions can be found at http://journals.plos.org/plosone/s/latex.

3.In your Data Availability statement, you have not specified where the minimal data set underlying the results described in your manuscript can be found. PLOS defines a study's minimal data set as the underlying data used to reach the conclusions drawn in the manuscript and any additional data required to replicate the reported study findings in their entirety. All PLOS journals require that the minimal data set be made fully available. For more information about our data policy, please see http://journals.plos.org/plosone/s/data-availability.

Reviewers' comments:

Reviewer's Responses to Questions

**Comments to the Author**

1. Is the manuscript technically sound, and do the data support the conclusions?

Reviewer #1: Partly

Reviewer #2: Yes

2. Has the statistical analysis been performed appropriately and rigorously? 

Reviewer #1: I Don't Know

Reviewer #2: Yes

3. Have the authors made all data underlying the findings in their manuscript fully available?

Reviewer #1: No

Reviewer #2: Yes

4. Is the manuscript presented in an intelligible fashion and written in standard English?

Reviewer #1: Yes

Reviewer #2: Yes

5. Review Comments to the Author

Reviewer #1: Please see attachment for comments. I did not see any data accompanying the manuscript. Because data was not provided, it is unclear if the statistical analysis was performed appropriately and rigorously.

Reviewer #2: The paper is well written. As noted by the researchers, there are very few quantitative studies on the relationship between religion and young trans women. Thus the study contributes to the literature. Please see my comments below.

1. Does religious affiliation play a role?

Some affiliations are more welcoming than others. Does your database have a variable for religious affiliation that you can also include in the model? This would allow you to see if your findings on the impact of personal religiosity vary by religious affiliation. For instance, you can interact the religious affiliation variable with the religious importance variable and use some effect plots to study this. If your database does not allow for this, then this marks another limitation of your analysis.

2. Selection effects

The study assumes an exogenous relationship between religiosity and anti-gay attitudes. A religious institution’s stance on transgenderism (that is gender-affirming or not) could influence where transwomen worship and the importance they place on religion in their lives. The observed relationship could be due to selection effects, which marks a significant limitation of the findings that should be noted in the study.

6. PLOS authors have the option to publish the peer review history of their article (what does this mean?). If published, this will include your full peer review and any attached files.

Reviewer #1: No

Reviewer #2: No

---

## [Author Response · Author response to Decision Letter 0]

20 Nov 2022

Major Revisions:

Reviewer 1

Comment 1: Since this is a cross-sectional study, cause and effect relationships cannot be

determined. Therefore, religiosity may cause resilience OR resilience may cause

religiosity OR both religiosity and resilience may be caused by a third or additional

factors. The authors keep framing the association in this study as if religiosity causes

resilience, in both the Introduction and Discussion sections. The authors should also

consider and articulate other cause-and-effect relationships in both the Introduction

and Discussion sections.

For example, consider this scenario. This study was conducted in the SF Bay

Area, a very wealthy area marked by liberal and progressive city and state

policies that may be well-funded. Could it be that this economic/political

environment promotes resilience in a variety of ways? And thus, already-resilient

people are more likely to participate in social/civil society. And for Black young

trans women (YTW), this greater participation may take the form of involvement

in churches, hence, scoring high on religiosity.

Response 1: 

Thank you for this comment. We agree that cause and effect cannot be established with cross-sectional data. As suggested, we have included additional text in the Introduction and Discussion sections to elaborate on the potential impact that resilience can have on religiosity on pages 5 and 22 (additions are highlighted below).

Page 5: “Second, this study is cross-sectional, limiting our ability to determine the temporal nature of the relationship between religiosity and resilience; namely, that being more religious causes increased resilience. It is also possible that individuals with greater resilience are more likely to participate in civil society, including in religious organizations, thus leading to increased religiosity.”

Page: 22 “In understanding the relationship between religiosity and resilience, it is also important to note that the relationship between religiosity and resilience is not necessarily unidirectional. A reciprocal relationship can exist, in which resilience enables greater civic engagement, including participation in religious practices and organizations.”

Comment 2: In the Introduction section, the conceptual/theoretical construct for examining the

possible association between religiosity and resilience is weak. Religiosity has a

psychological component but is also social and spiritual. Social support is barely

mentioned and it would behoove the authors to further unpack how social support is

fundamental to, and perhaps may even mediate, the association between religiosity and

resilience.

If the authors choose to further unpack and discuss social support, then it would

also be useful to discuss how social support can build social capital with respect

to cultivating resilience. A landmark text on social capital is Putnam’s (1995,

2000) work on bonding and bridging social capital. Hwahng et al., 2021, have

discussed how social capital is related to resilience among trans women of color

and have taken a more granular focus of “trust capital” within social capital, i.e.

“thick trust” within bonding social capital and “thin trust” within bridging social

capital. Given the intimate relationships (to a divine being and/or other

members of a given faith community) inherent in religiosity, discussion of “trust

capital” may be particularly salient to this article’s premise. It would also be

useful to include additional literature on social support and social capital within

transgender communities such as Perez-Brumer et al., 2017; Erosheva et al.,

2016; and Pinto et al., 2008. Finally, incorporating the “space of capabilities”

construct (Sen, 1999) may also be useful since space of capabilities has

implications for both resilience as well as religious practice.

Response 2:

Thank you for these recommendations. In the introduction section on pages 5-7, with additional review of the literature, we have included discussion on the pathways through which religiosity can influence resilience. We discuss the individual coping resources that some obtain through religiosity. We further discuss the expansion of social networks through participation in religious communities and highlight the potential impacts on social trust and social capital. Finally, we incorporate Sen’s “capability approach” to illustrate the impact that resilience and religious practice can have on individual wellbeing. To support these points, we have incorporated the reviewer’s suggested sources.

Pages 5-7: “Religiosity has been demonstrated to promote resilience through multiple pathways. Religiosity provides some people with coping resources such as benevolent religious reappraisal (i.e., finding positive meaning from an adverse event), coping self-efficacy (i.e., confidence in one’s ability to cope with adverse events), collaborative problem solving with a higher being, and seeking spiritual support from a higher being. (22,23)

Additionally, religiosity provides some with access to religious social networks that bolster resilience. Participation in religious organizations allows individuals to develop relationships with others who share similar spiritual and religious beliefs.(24) These social connections contribute to social capital, a group-level construct that includes networks, norms, and trust, which promotes coordinated action, and subsequently, collective access to social and material resources.(25) 

Social trust is a cognitive domain of social capital which is strengthened by social interactions and allows groups of people to work together for a common goal. (26) Two types of social trust are theorized to exist. Thick trust refers to trust that is built on close relationships between individuals with shared social identity (e.g., family and friends) and thin trust refers to trust that is built on looser relationships with those of different social identities.(26) Thin trust is often developed through participation in formal institutions (e.g., community organizations and work).(26) 

Research with trans women of color has revealed that thick trust builds social capital within groups by allowing members to receive emotional support from each other, express themselves authentically to each other, and receive tangible support (e.g., social, physical, and financial) from one another.(27) With a strong foundation of thick trust, some trans women feel empowered to accept and embrace differences of individuals within and outside of their groups, allowing them to build greater thin trust.(27) Thin trust enables some trans women to access resources previously unfamiliar to them and to work with organizations and other groups to expand access to medical, legal, and employment resources through advocacy and political action.(27) 

Participation in religious communities and organizations provides individuals with access to diverse social networks that are bound together by a common spiritual identity, thus potentiating the development of thin and thick trust.(28) Ultimately, social trust fostered through participation in religious communities can facilitate development of social capital and greater resilience through increased access to social and physical resources. 

Social capital has been demonstrated to bolster the resilience of trans women in a variety of settings. Trans women living with HIV have leveraged social capital within their communities to collaboratively overcome healthcare barriers, improve access to community health resources, and shift public stigma and discrimination.(29) Black and Latina transgender women have reported utilizing social capital to acquire emotional support, access gender affirming medical care, and organize civic and political action.(30) These resources, made available by social capital, help trans women overcome challenges generated by societal stigma and discrimination.

Resilience and participation in religious activities can independently contribute to an individual’s overall sense of wellbeing. This relationship can be understood through the “capability approach”, which details that wellbeing is dependent on an individual’s perceived ability to perform functions and activities that they value.(31) Inherent to the performance of most activities is some level of risk. Therefore, an individual’s resilience can help them to expand their capabilities by allowing them to overcome fear and bounce back from failure in pursuit of performing desired life functions.(32) 

Under the framework of the “capability approach”, participation in religious and spiritual activities can represent an important function among those who value religion. Therefore, freedom to participate in religious communities and organizations as an individual desires can promote wellbeing.(33) Conversely, barriers to spiritual and religious practice, which many transgender people face, may be detrimental to wellbeing if religious practice is central to their cultural identity or value system.”

Comment 3: The authors keep referring to “personal religiosity” in the Introduction and Discussion sections. Yet in the limitations the authors admit that the collected data was based on a question about the “importance of religion” in their lives and it is unknown if this importance is due to participation in faith-based organizations and communities or “personal spirituality.” And participation in faith-based organizations and communities could be characterized as social religiosity, cultural religiosity, and/or collective religiosity, which is distinct from “personal religiosity.” The authors thus need to clean up this language in the Introduction and Discussion sections and clearly state the actual research question that was asked in those sections, which is about the importance of religion, and not per se about “personal religiosity.”

Response 3:

Thank you for this comment. We have modified text in the Introduction and Discussion sections on pages 8 and 19 to prevent conflation of “personal religiosity” with “religiosity” and the “importance of religion”. We consistently use “importance of religion” when referring to our data.

Page 8: “The primary goal of this study is to examine the association between the importance of religion and resilience among young trans women to determine if greater importance of religion could be a resilience promoting factor. The secondary goal is to identify demographic factors that are associated with importance of religion among young trans women.”

Page 19: “Our finding is consistent with qualitative research reporting that despite stigma from religious institutions, some transgender individuals identify religiosity to be a vital source of resilience.(45,46) Specifically, transgender participants from prior research have reported that their religiosity helps them to cope with adversity, foster hope, accept their personal identity, find their chosen family, and heal past trauma.(39–42,46)”

Comment 4: Given that the importance of religion for YTW may be due to past and/or current

participation in faith-based organizations, it would be useful to conduct a more

in-depth literature search on LGBTQ-inclusive and friendly faith-based

organizations (including churches, mosques, temples, synagogues, etc.) and

write at least one paragraph discussing these religious resources.

Response 4:

Thank you for this recommendation. On pages 21-22, we have included an in-depth review of existing LGBTQ inclusive religious organizations that support transgender individuals.

Pages 21-22: “Given the positive association between religiosity and resilience, efforts to support the spiritual needs of trans women may help to foster resilience. The importance of religion in the lives of some young trans women may stem from past or current participation in faith-based organizations. Within the United States, various LGBTQ-affirming places of worship exist. Mainstream religious denominations including the Unitarian Universalist Association, the United Church of Christ, the Episcopal church, Reform and Conservative Judaism, the Evangelical Lutheran Church in America, and the Presbyterian Church (USA) have officially adopted LGBTQ-inclusive policies.(47) Glide Memorial Church, in San Francisco, California has a long history of LGBTQ inclusion and advocacy and has hosted a Gender Expression & Identity summit, with the goal of fostering solidarity with the trans community and to hold space for trans individuals to express their spirituality.(48) Masjid al-Rabia is a LGBTQ-affirming Islamic community center in Chicago, Illinois that facilitates support groups for trans and gender nonconforming Muslims.(49) The Brooklyn Zen Center in Brooklyn, New York creates space for trans Buddhists, by hosting events that highlight the voices of trans Buddhist leaders.(50) While not comprehensive, these gender affirming faith-based organizations represent some of the religious communities that transgender people may participate in. Effort should be made to expand and support faith-based organizations like these, which serve as promoters of social capital and resilience for trans individuals. Additionally, transphobic doctrine and attitudes within all religious organizations must be reformed so that trans women can have greater access to this potential resilience promoting resource.”

Comment 5: The authors could also conduct a more comprehensive lit review on LGBTQ

people, resilience, and religion/spirituality. For example there has been a

systematic review on religion, spirituality, mental health, and LGBTQ youth that

was recently published (McCann et al., 2020) and a special issue on LGBTQ

studies in Psychology of Religion and Spirituality (Yarhouse et al., 2021) that

were not discussed.

Response 5:

Thank you for your referral to these sources. We carefully read the reviews and their primary source publications. We found none of the papers included in the McCann et al., 2022 review included transgender participants. Only one of the papers included in the Yarhouse et al., 2021 review includes transgender participants. These findings confirm the gap in the literature and the need for research specifically for transgender persons. While there are sometimes similarities between the experiences of sexual and gender minority youth, we understand that the experiences of transgender youth are often dissimilar to those of lesbian, gay, and bisexual youth. For example, even within LGB-affirming faith-based organizations, some transgender people have reported exclusion and discrimination due to their gender identity. As mentioned above, one of the papers (Gandy et al. 2021) in the Yarhouse et al. 2021 review includes interviews with trans youth. We have included discussion of the findings of this paper in the introduction section on page 8. 

Page 8: “Some transgender people find their chosen family in supportive religious communities after facing rejection from their family of origin.(39) Furthermore, belief in a divine creator helps some transgender people to embrace and accept their identity.(40) In these supportive religious environments, transgender individuals report experiencing healing from past rejection and loss, inclusion, belonging, and joy.(39)” 

Comment 6: It is unclear what the applications of this study may be, beyond just conducting more research. Are the authors interested in examining what factors may promote greater resilience for YTW? Or do the authors want barriers to be removed so that more YTW can be involved in faith-based organizations? Or are the authors interested in YTW participating in greater civil society, more broadly defined? It would be useful for the reader to know what the overall intentions of the authors are in conducting this study, especially since causality cannot be established.

Response 6:

Thank you for encouraging us to more boldly discuss the applications of our research. Our study highlights religiosity as a resilience resource for young trans youth. Based on this finding, we believe that support for existing trans-affirming faith-based organizations and the development of new affirming faith-based organizations that cater to the needs of trans women can promote resilience. Additionally, our findings support efforts to remove barriers to participation in existing faith-based organizations via reforming transphobic doctrine and attitudes. Application of study findings has been added to the Discussion section on page 22. 

Page 22: “Effort should be made to expand and support faith-based organizations like these, which serve as promoters of social capital and resilience for trans individuals. Additionally, transphobic doctrine and attitudes within all religious organizations must be reformed so that trans women can have greater access to this potential resilience promoting resource.”

Minor Revisions:

Comment 1: On page 8, change heading from “Dependent Variable” to “Outcome of Interest” or another heading. This article is not merely a statistical exercise.

Response 1: The heading has been changed to “Outcome of Interest” on page 11.

Comment 2: For Table 1, it would be useful to break down the categories of “Education” and

“Monthly Income” by gender identity and race. For example, it might be of interest to

note if certain racial or gender groups are more likely to have completed some college

or have higher income compared to other racial or gender groups.

Response 2: Thank you for this suggestion. We assessed the differences in demographic characteristics by gender identity and race/ethnicity and found that college education was higher among White compared to racial/ethnicity minority trans youth. We have incorporated this finding into the text. No other differences were found.

Comment 3: In Table 1, more space has to be included to separate the “Age” row from the rest of the table since the columns represent different measures for the “Age” row compared to

the rest of the table.

Response 3: More space has been added after Age in Table 1 on page 13.

Comment 4: In Tables 2 and 3, list adjusted variables at the bottom of these tables.

Response 4: We have included the variables adjusted for in the footnotes of Tables 2 and 3 on pages 16 and 18.

Reviewer 2

Comment 1: Does religious affiliation play a role? Some affiliations are more welcoming than others. Does your database have a variable for religious affiliation that you can also include in the model? This would allow you to see if your findings on the impact of personal religiosity vary by religious affiliation. For instance, you can interact the religious affiliation variable with the religious importance variable and use some effect plots to study this. If your database does not allow for this, then this marks another limitation of your analysis.

Response 1: Thank you for this important question. Our survey did not collect information on the religious affiliation of participants. We recognize that religious affiliation may impact the way that religiosity influences resilience. Future studies should collect information on participant religious affiliation to better understand how it influences the relationship between religiosity and resilience. We have included this as a limitation of the study on page 23.

Page 23: “Fourth, information on religious affiliation with respect to religion, denomination, and degree of trans-affirmation was not collected from participants. The degree of trans-affirmation within a religious organization’s policies and culture may influence the effects that religiosity has on resilience of transgender members. Therefore, lack of consideration of religious affiliation of participants represents a limitation of this study.”

Comment 2: Selection effects. The study assumes an exogenous relationship between religiosity and anti-gay attitudes. A religious institution’s stance on transgenderism (that is gender-affirming or not) could influence where transwomen worship and the importance they place on religion in their lives. The observed relationship could be due to selection effects, which marks a significant limitation of the findings that should be noted in the study.

Response 2: Thank you for this observation. We recognize that the trans-affirmativeness of faith communities that trans women are a part of may be a confounder in the relationship between religiosity and resilience. We have included discussion of this in the limitations section on page 23.

Page 23: “Additionally, the degree of trans-affirmation of faith-based organizations that trans women participated in may independently influence their religiosity and resilience. Therefore, the observed association between religiosity and resilience within this study could theoretically be attributed to the confounding effects of the trans-affirmativeness of faith communities that participants were a part of. Future studies may examine the relationships between religiosity and resilience while controlling for trans-affirmativeness of faith-based communities that participants are members of.”

---

## [Decision Letter · Decision Letter 1]

26 May 2023

PONE-D-22-01853R1The association between religiosity and resilience among young trans womenPLOS ONE

Dear Dr. Wang,

Thank you for submitting your manuscript to PLOS ONE. After careful consideration, we feel that it has merit but does not fully meet PLOS ONE’s publication criteria as it currently stands. Therefore, we invite you to submit a revised version of the manuscript that addresses the points raised during the review process.

We look forward to receiving your revised manuscript.

Kind regards,

Jianhong Zhou

Staff Editor

PLOS ONE

Journal Requirements:

Reviewers' comments:

Reviewer's Responses to Questions

**Comments to the Author**

1. If the authors have adequately addressed your comments raised in a previous round of review and you feel that this manuscript is now acceptable for publication, you may indicate that here to bypass the “Comments to the Author” section, enter your conflict of interest statement in the “Confidential to Editor” section, and submit your "Accept" recommendation.

Reviewer #1: (No Response)

Reviewer #2: (No Response)

2. Is the manuscript technically sound, and do the data support the conclusions?

Reviewer #1: Yes

Reviewer #2: Yes

3. Has the statistical analysis been performed appropriately and rigorously? 

Reviewer #1: Yes

Reviewer #2: N/A

4. Have the authors made all data underlying the findings in their manuscript fully available?

Reviewer #1: No

Reviewer #2: No

5. Is the manuscript presented in an intelligible fashion and written in standard English?

Reviewer #1: Yes

Reviewer #2: Yes

6. Review Comments to the Author

Reviewer #1: This revised version is a great improvement from the original version and the authors are commended for incorporating the feedback from the reviewers. A few minor revisions are suggested.

Please see the attached document with review comments to the author.

Reviewer #2: I would like to congratulate the author for adequately responding to my comments. With the changes made, I believe the paper is now acceptable for publication

7. PLOS authors have the option to publish the peer review history of their article (what does this mean?). If published, this will include your full peer review and any attached files.

Reviewer #1: No

Reviewer #2: No

---

## [Author Response · Author response to Decision Letter 1]

21 Jun 2023

1. There seems to be inconsistency throughout the article of either using “Black/African

American” or “Black” to refer to the same racial identity, especially in the Discussion

section. Please use only one of these racial identity labels consistently throughout the

article.

Thank you for this recommendation. To maintain consistency, we have decided to utilize “Black/African American” as this is the current official term used by the San Francisco Department of Public Health. This has been modified throughout the manuscript.

2. In discussing the Results of Table 3 on page 16, it should also be mentioned that those

who identified as “Other” reported less resilience compared to those who identified as

‘Female,” since this is a significant finding.

Thank you for this suggestion. We have included mention of “other” gender identity when discussing factors associated with resilience:

 “Black/African American (aOR 4.25, 95% CI 1.56, 11.62, p=.005) and Hispanic/Latinx race/ethnicity (aOR 2.49, 95% 1.30, 4.77, p=.006), ‘other’ gender identity (aOR .29, 95% CI .09, .95, p=.04), and monthly income >$1,000 (aOR 3.87, 95% CI 2.00, 7.50, p=<.001) were also independently associated with increased resilience.”

3. On page 20, lines 323-327, it is odd to briefly discuss that higher income was associated

with increased resilience, since this association is not central to your research question.

And it is unclear why the higher income association is discussed yet Hispanic/Latinx

identity as a resilience association is not discussed in the Discussion section. Please

either discuss both variables or neither variable.

Thank you for this recommendation. We agree that the discussion of income is not central to our research aim. As such, we opted for your latter suggestion and have removed the paragraph mentioning income in the discussion section.

4. On page 20, line 338, please write “an LGBTQ-affirming” instead of “a LGBTQ-affirming”

This change has been made in the manuscript.

5. Given what is now written in the Introduction section on social capital fostering

resilience, it seems that “social capital” could also be considered as a possible mediator.

2

Thus, “social capital” could be inserted on page 21, lines 350-351, before social support

so that it now reads “(e.g., increased social capital, social support…”

Thank you for this recommendation. We have included social capital as a potential mediator: “Future studies are needed to identify factors that mediate the relationship between religiosity and resilience among transgender youth (e.g., increased social capital, social support, positive outlook, well-being)” 

6. “Hope” and “acceptance” on page 21, line 351 are vernacular terms that are not very

much recognized scientifically. Can these terms be replaced with more scientifically

recognized terms such as “well-being” and/or “positive outlook”?

Thank you for this suggestion. We have modified the manuscript as follows: “Future studies are needed to identify factors that mediate the relationship between religiosity and resilience among transgender youth (e.g., increased social capital, social support, positive outlook, well-being)”

---

## [Decision Letter · Decision Letter 2]

7 Jul 2023

The association between religiosity and resilience among young trans women

PONE-D-22-01853R2

Dear Dr. Wang,

We’re pleased to inform you that your manuscript has been judged scientifically suitable for publication and will be formally accepted for publication once it meets all outstanding technical requirements.

Kind regards,

Thomas E. Guadamuz, Ph.D., M.H.S.

Academic Editor

PLOS ONE

Additional Editor Comments (optional):

Dear authors,

I have recommended acceptance of this paper contingent on being clear in the text with respect to the "other" gender identity of whether this is associated with increased or decreased resilience as reviewer 1 has noted.

Best,

Thomas Guadamuz

Reviewers' comments:

Reviewer's Responses to Questions

**Comments to the Author**

1. If the authors have adequately addressed your comments raised in a previous round of review and you feel that this manuscript is now acceptable for publication, you may indicate that here to bypass the “Comments to the Author” section, enter your conflict of interest statement in the “Confidential to Editor” section, and submit your "Accept" recommendation.

Reviewer #1: (No Response)

Reviewer #2: All comments have been addressed

2. Is the manuscript technically sound, and do the data support the conclusions?

Reviewer #1: Partly

Reviewer #2: Yes

3. Has the statistical analysis been performed appropriately and rigorously? 

Reviewer #1: I Don't Know

Reviewer #2: Yes

4. Have the authors made all data underlying the findings in their manuscript fully available?

Reviewer #1: No

Reviewer #2: Yes

5. Is the manuscript presented in an intelligible fashion and written in standard English?

Reviewer #1: Yes

Reviewer #2: Yes

6. Review Comments to the Author

Reviewer #1: Please see my uploaded Comment to Authors. Given the findings, "other" gender identity would be associated with decreased resilience in Table 3.

Reviewer #2: (No Response)

7. PLOS authors have the option to publish the peer review history of their article (what does this mean?). If published, this will include your full peer review and any attached files.

Reviewer #1: No

Reviewer #2: No

---

## [Editor Report · Acceptance letter]

24 Jul 2023

PONE-D-22-01853R2 

The association between religiosity and resilience among young trans women 

Dear Dr. Wang:

I'm pleased to inform you that your manuscript has been deemed suitable for publication in PLOS ONE. Congratulations! Your manuscript is now with our production department. 

Kind regards, 

on behalf of

Dr. Thomas E. Guadamuz 

Academic Editor

PLOS ONE